# Responsivity of Two Pea Genotypes to the Symbiosis with Rhizobia and Arbuscular Mycorrhiza Fungi—A Proteomics Aspect of the “Efficiency of Interactions with Beneficial Soil Microorganisms” Trait

**DOI:** 10.3390/ijms26020463

**Published:** 2025-01-08

**Authors:** Andrej Frolov, Julia Shumilina, Sarah Etemadi Afshar, Valeria Mashkina, Ekaterina Rhomanovskaya, Elena Lukasheva, Alexander Tsarev, Anton S. Sulima, Oksana Y. Shtark, Christian Ihling, Alena Soboleva, Igor A. Tikhonovich, Vladimir A. Zhukov

**Affiliations:** 1Laboratory of Analytical Biochemistry and Biotechnology, K.A. Timiryazev Institute of Plant Physiology Russian Academy of Science, 119334 Moscow, Russia; schumilina.u@yandex.ru (J.S.); oriselle@yandex.ru (A.S.); 2Institute of Pharmacy, Martin-Luther Universität Halle-Wittenberg, 06120 Halle, Germany; sarah.etemadiafshar94@gmail.com (S.E.A.); christian.ihling@pharmazie.uni-halle.de (C.I.); 3Faculty of Biology, St. Petersburg State University, 199034 St. Petersburg, Russia; lera.sleepy@gmail.com (V.M.); rcatherine@mail.ru (E.R.); elena_lukasheva@mail.ru (E.L.); alexandretsarev@gmail.com (A.T.); i.tikhonovich@arriam.ru (I.A.T.); 4All-Russia Research Institute for Agricultural Microbiology, 196608 St. Petersburg, Russia; asulima@arriam.ru (A.S.S.); oshtark@arriam.ru (O.Y.S.)

**Keywords:** legume–rhizobial symbiosis, arbuscular mycorrhiza, pea, efficiency of interactions with beneficial soil microorganisms, proteomics

## Abstract

It is well known that individual pea (*Pisum sativum* L.) cultivars differ in their symbiotic responsivity. This trait is typically manifested with an increase in seed weights, due to inoculation with rhizobial bacteria and arbuscular mycorrhizal fungi. The aim of this study was to characterize alterations in the root proteome of highly responsive pea genotype k-8274 plants and low responsive genotype k-3358 ones grown in non-sterile soil, which were associated with root colonization with rhizobial bacteria and arbuscular mycorrhizal fungi (in comparison to proteome shifts caused by soil supplementation with mineral nitrogen salts). Our results clearly indicate that supplementation of the soil with mineral nitrogen-containing salts switched the root proteome of both genotypes to assimilation of the available nitrogen, whereas the processes associated with nitrogen fixation were suppressed. Surprisingly, inoculation with rhizobial bacteria had only a minor effect on the root proteomes of both genotypes. The most pronounced response was observed for the highly responsive k-8274 genotype inoculated simultaneously with rhizobial bacteria and arbuscular mycorrhizal fungi. This response involved activation of the proteins related to redox metabolism and suppression of excessive nodule formation. In turn, the low responsive genotype k-3358 demonstrated a pronounced inoculation-induced suppression of protein metabolism and enhanced diverse defense reactions in pea roots under the same soil conditions. The results of the study shed light on the molecular basis of differential symbiotic responsivity in different pea cultivars. The raw data are available in the PRIDE repository under the project accession number PXD058701 and project DOI 10.6019/PXD058701.

## 1. Introduction

Mineral fertilizers represent an essential component of global agriculture, but their overuse can negatively affect soil health and biodiversity [1]. Currently, attempts are being made to reduce the use of mineral fertilizers, while maintaining the same yields and quality of the final agricultural product [2]. Legumes (Fabaceae) are of particular interest in this regard, as they can be independent from mineral nitrogen sources, due to mutualistic endosymbiosis with nitrogen-fixing nodule bacteria (rhizobia) and arbuscular mycorrhizal (AM) fungi of the phylum Glomeromycota [3,4]. In terms of these plant–microbial interactions, rhizobia supply the plant with metabolically accessible nitrogen, whereas AM fungi provide water and hardly soluble phosphates. On the other hand, the microorganisms gain access to plant photosynthetic assimilates and find a well-balanced habitat within the plant rhizosphere. 

Multiple studies have confirmed that these symbioses can improve the overall fitness of the plant and, ultimately, positively impact on seed productivity and quality [5,6]. However, the overall outcomes of symbiotic relationships are not predetermined and depend on multiple factors, in particular on the genotypes of plant and microorganism partners [7,8,9]. Thus, it is well known that different pea genotypes exhibit different responsivity to inoculation with rhizobia and/or AM fungi. This diversity in plant response to inoculation is underlined with a genetic trait known as EIBSM (efficiency of interactions with beneficial soil microorganisms) and manifested as an increase in the seed biomass productivity (seed weights) upon inoculation with rhizobia and/or AM fungi [10,11]. In agreement with this, ‘responsive’ and ‘non-responsive’ genotypes have been described in the pea (*Pisum sativum* L.). This responsivity trait was manifested with high or low gain of seed weight upon inoculation [12,13]. Despite the fact that the molecular mechanisms behind the establishment and functioning of the symbioses in legumes are well studied [14,15,16], only a little information about the molecular mechanisms underlying the plant responsivity to inoculation is available [13]. Filling this knowledge gap would give access to powerful tools for marker-assisted breeding and genomic selection to establish new legume cultivars with high symbiotic responsivity.

The pea (*Pisum sativum* L.) is recognized worldwide as one of the most important crops (FAOSTAT 2021). Besides, it represents a convenient model to address the mechanisms behind establishment and functioning of the nitrogen-fixing symbiosis and arbuscular mycorrhiza. Recent achievements in the field of pea genomics and high throughput transcriptomics and proteomics make it possible to uncover the molecular bases of the traits of interest [17], including the EIBSM trait [13,18]. Our earlier proteomics study, accomplished with the seeds of two pea genotypes with high and low responsivity to symbiosis, showed that the high responsivity to combined inoculation with rhizobia and AM fungi was associated with prolongation of the seed filling period [18]. The aim of the present work was to investigate the molecular mechanisms of the EIBSM trait by description of the proteomic changes in the roots of pea genotypes, the ‘responsive’ k-8274 and the ‘non-responsive’ k-3358, after inoculation with nodule bacteria (NB), nodule bacteria plus AM fungi (NB+AMF), and under mineral nutrition (MN), as compared to a non-treated control.

## 2. Results

### 2.1. Isolationof Proteins and Tryptic Digestion

To ensure efficient extraction of proteins and the maximal possible coverage of the pea proteome, we used the method of phenol extraction for isolation of the total protein fraction. Determination of the protein concentrations in the obtained isolates revealed extraction yields in the range of 0.9–3.5 mg/g fresh weight (Appendix A). The SDS-PAGE revealed overall lane densities of 1.5 *×* 10^4^–1.9 *×* 10^4^ arbitrary units (AU), yielding the relative standard deviation (RSD) of 6.9% (Appendix A). This value was sufficient for sample amount normalization and did not require additional recalculation of the digest loads on the LC column.

The signal patterns observed in the electropherograms showed low intra-group variability (i.e., dispersion within one group of plants grown under the same soil conditions) and were similar between the lanes corresponding to different treatment groups (Appendix A). Tryptic digestion of the obtained protein isolates was considered to be complete, as no bands could be detected in the electropherograms of the corresponding hydrolysates acquired with the same load (Appendix A).

### 2.2. Protein Annotation

In total, 5382 and 5356 peptides were identified by the MS/MS-based database search in the datasets acquired for the *P. sativum* genotypes k-8274 and k-3358, respectively (Appendix A). Among these, for the k-8274 genotype, 2003 peptides occurred in all four groups, while 699, 199, 233 and 377 peptides were unique for the control plants (C), those grown in the presence of rhizobial symbiosis (nodule bacteria, NB), in the presence of NB and arbuscular mycorrhiza (NB+AMF) and in the presence of mineral nitrogen source in soil without inoculation with microorganisms (mineral nutrition, MN), respectively. Analogously, in total, 2167 peptides were identified in all experimental plants of the k-3358 genotype, while 233, 343, 294 and 575 peptides were unique for the C, NB, NB+AMF and MN groups, respectively.

Based on these identifications, altogether, 2606 possible proteins (2237, 1746, 1637 and 1899 in the C, NB, NB+AMF and MN groups) could be annotated for the k-8274 genotype (Appendix A). They represented, altogether, 1348 non-redundant proteins (protein groups), with 1138, 869, 821 and 940 annotated to the controls, NB, AMF+NB and MN groups, respectively (Figure 1a, Appendix A). Among these, 1281 non-redundant proteins were common for all four groups, while 312, 75, 75 and 128 proteins were unique for controls, NB, NB+AMF and MN groups, respectively. Analogously, in total, 2570 probable proteins could be annotated in the plants of the k-3358 genotype, with 1699, 1948, 1866 and 2062 accessions annotated to the controls, NB, NB+AMF and MN groups, respectively (Appendix A). Therefore, 1327 proteins were common for the all four groups, while 89, 111, 108 and 241 polypeptides were unique for the controls, NB, NB+AMF and MN groups, respectively (Appendix A). These annotations represented, altogether, 1331 protein groups, 887, 977, 969 and 1008 of which were identified in the controls, NB, NB+AMF and MN groups, respectively (Figure 1b, Appendix A). Therefore, 89, 111, 108 and 241 non-redundant proteins were unique in the controls, NB, NB+AMF and MN groups, respectively.

It is important to note that the MS/MS-based assignment of individual proteins as unique totally relies on confident prediction of the peptide sequences without any consideration of the inter-group alignments at the MS1 level. Obviously, such alignments (typically employed as a part of the label-free quantification (LFQ) protocol), would, on one hand, reduce the numbers of treatment-specific unique proteins (i.e., identification of individual proteins will be independent from the growing conditions); on the other hand, they would allow precise assessment of the inter-group abundance differences for individual “common” proteins. Therefore, this strategy was employed as the next step.

### 2.3. Label-Free Relative Quantification of the Annotated Proteins

To obtain better insight in the molecular mechanisms underlying the inter-line differences in the responses to symbiotic microorganisms and supplemented mineral nutrition, we decided to address this question in a series of paired inter- and intra-genotype comparisons. Specifically, we compared the protein expression patterns in the roots of the control k-8274 and k-3358 plants, along with three intra-group comparisons (MN vs. control, NB vs. control and AMF+NB vs. control) for each of the two genotypes. For this, the *t*-test was employed with the cut-off criteria FC (protein abundance fold change) ≥ 1.5 and *p*_adjusted_ ≤ 0.05. In these seven paired comparisons, altogether, 491 differentially abundant proteins could be found. The top three up- and down-regulated proteins identified in each comparison, which appeared to be the most responsive to different soil conditions (i.e., legume–rhizobial symbiosis, arbuscular mycorrhiza and supplementation with mineral sources of nitrogen vs. control plants grown in the absence of soil supplementation), are listed in Table 1. The complete patterns of differential protein expression are described in more detail below.

Thus, 44 proteins could be identified as differentially abundant, when the genotypes k-8274 and k-3358 were compared in the absence of microorganism inoculations and soil supplementation with mineral nitrogen salts. Twenty-four and 20 proteins were more abundant in the roots of the k-8274 and k-3358 plants, respectively (Appendix A). The dirigent protein and histone 2B showed the most pronounced inter-genotype differences (5.3- and 3.7-fold, respectively) and were more represented in the roots of k-8274 plants, whereas a representative of the peptidase C1 family (Q40993), isoforms X1 (G7JCJ5) and MYB-CC type transcription factor were more abundant in the roots of k-3358 (34.8-, 18.4- and 16-fold inter-genotype differences, respectively).

Soil supplementation with mineral nitrogen salts resulted in different responses of the genotypes k-8274 and k-3358 to external supplementation with this vital macronutrient. Thus, the k-3358 plants appeared to be more responsive to these conditions, with 179 differentially expressed proteins represented, 98 and 81 of which were up- and down-regulated (in comparison to the corresponding controls), respectively (Appendix A). The up-regulation pattern was dominated with histone H2B, protein G7JCJ5 isoform X1 and Luc7-like protein (11.4-, 5.2- and 4.9-fold, respectively). Although the down-regulation pattern was less rich, it was featured with a much more pronounced group of symbiosis-related proteins, with associated enzymatic and regulatory machinery the most affected. The top responsive proteins of this group were the heat shock protein DnaJ (A2Q3G0), leghemoglobin K, carbonic anhydrase EC4.2.1.1 and leghemoglobin B, with 1390.8-, 150.9-, 129.2- and 98.9-fold decreases in relative abundances, respectively.

The genotype k-8274 showed a clearly less pronounced response to the soil supplementation with external nitrogen. Only 111 proteins appeared to be affected, with 55 and 56 up- and down-regulated accessions, respectively (Appendix A). The up-regulation pattern was different from the counterpart genotype and dominated with the uncharacterized protein LOC101501292 (A0A1S2YAH3), mono-copper oxidase SKS1-like protein (putative *L*-ascorbate oxidase, EC 1.10.3.3) and lipoxygenase (EC 1.13.11. Q14ST8) showing 9.0-, 6.4- and 4.1-fold relative abundance shifts (Appendix A). The down-regulation pattern showed less pronounced decreases in abundancies of the symbiosis-related proteins (Appendix A). Moreover, not only these proteins were in the most affected group. Indeed, β-fructofuranosidase cell wall isozyme 2-like protein (A0A1U9X403), unknown protein and putative chromatin CoxE-like regulator PHD family were the most regulated (246.3, 153.9 and 121.1-fold, respectively).

When the k-8274 and k-3358 plants were inoculated with rhizobia, the proteome response appeared to be principally different. Thus, the proteome of the former k-3358 genotype did not respond to the inoculation with rhizobia, i.e., no changes in protein dynamics could be observed when the plants of the k-8274 genotype were inoculated with the microorganism culture. In contrast, 13 and ten proteins were identified as up- and down-regulated when rhizobia colonized the roots of the k-3358 plants (Appendix A). The 60S ribosomal protein L5-2, protein G7JCJ5 isoform X1 and glutelin type-B-like protein (putative 11-S seed storage protein) most strongly contributed to the up-regulation pattern, with 3.9-, 3.2- and 2.5-fold alterations, respectively (Appendix A). Although the down-regulation pattern included only ten proteins, it generally resembled the response of this genotype to soil supplementation with external nitrogen, with CBS/octicosapeptide/phox/Bemp1 (PB1) domain protein, putative serine/threonine-protein kinase (A0A392LZA9) and leghemoglobin most strongly affected (16.4-, 2.4- and 2.3-fold, respectively, Appendix A).

Finally, combined inoculation of the k-3358 and k-8274 plants with arbuscular mycorrhiza fungi and rhizobia (AMF+NB) yielded clearly different responsivity of the corresponding root proteomes (Appendix A). Thus, the former genotype responded to the combined inoculation with 16 up- and 17 down-regulated proteins. The uncharacterized protein A0A2Z6PHW6, caffeic acid *O*-methyltransferase (belongs to putative methyltransferases EC 2.1.1.) and histone H2B.1 appeared to be the most strongly up-regulated species (3.5-, 2.8- and 2.5-fold, Appendix A), whereas putative wall-associated serine/threonine-protein kinase (A0A392LZA9), uncharacterized protein A0A396ITY0 and carboxypeptidase EC 3.4.16.- (G7IT50) were the most down-regulated proteins (3.6- 3.4- and 3.3-fold, respectively, Appendix A).

In comparison to the genotype k-3358, the root proteome of the k-8274 plants proved to be much more responsive to the combined inoculation with two microorganisms (AMF+NB). This response was manifested with 39 up- and 59 down-regulated proteins (Appendix A). The monocopper oxidase SKS1-like protein (putative *L*-ascorbate oxidase EC 1.10.3.3), HTH myb-type domain-containing protein (MYB-CC type transcription factor) and mitochondrial-like arginase 1 showed the highest increase in relative abundancies (5.5-, 4.2- and 3.4-fold). The down-regulation pattern was much more represented (in terms of the numbers of affected proteins) with the heat shock protein DnaJ, early nodulin ENOD18 and putative wall-associated serine/threonine-protein kinase as the most strongly affected accessions (67.8-, 44.9- and 26.3-fold, respectively).

### 2.4. Functional Annotation of Differentially Expressed Proteins

For the functional annotation of the differentially expressed *P. sativum* proteome, Mercator 4 v.2 software was used. The annotated proteins represented, altogether, 31 functional classes (bins) among the 35 available (Figure 2 and Figure 3, Appendix A).

Comparison of the root proteomes obtained from the control k-8274 and k-3358 plants revealed 14 and 13 functional classes, respectively. Most of these classes were represented by one or two proteins (Figure 2a, Appendix A). Among the proteins, more abundant in the roots of the genotype k-8274, the polypeptides involved in redox and secondary metabolism were the best represented (seven and three accessions, respectively, Appendix A). These groups included three peroxidases, two cytochromes b5, non-symbiotic leghemoglobin, and three proteins involved in polymerization of lignins in cell walls. In agreement with this, the cell wall (along with the group of secreted polypeptides) appeared to be one of the major predicted compartments for localization of the proteins, which were more abundant in the roots of k-8274 (Figure 2b).

On the other hand, bins 20 (stress), 26 (enzymatic families), 29 (protein metabolism) and 30 (signaling) were the most represented among the proteins, which showed higher relative abundance in the roots of the k-3358 plants (Appendix A). The first group included a lipoxygenase PLAT-plant-stress protein and a late embryogenesis abundant protein, while the second one comprised three enzymes involved in secondary metabolism—two methyl transferases and a representative of the dienelactone hydrolase family. Bin 29 was represented by cysteine peptidase C1 and serine carboxypeptidase S10, which are known to be involved in the regulation of plant development, seed maturation, senescence, and stress response [18,20,21]. Finally, the group of the proteins involved in signaling was represented by the RAB GTPase-like protein A1D (Ras-related protein Rab7).

Soil supplementation with mineral nitrogen induced a strong proteome response in the roots of the k-3358 plants. This response was manifested as pronounced up-regulation of 24 and down-regulation of 25 functional classes (Appendix A). The former was dominated with the polypeptides involved in protein metabolism, secondary and redox metabolism, as well as those annotated as “miscellaneous enzyme families”, with 18, 12, 10, and 7 accessions, respectively (Figure 3a, Appendix A).

The up-regulated part of the protein metabolism-related proteome is strongly dominated by its catabolic (degradation-related) component, including peptidases, proteases, and elements of the proteasomal degradation machinery. The up-regulation pattern of bin 16 (secondary metabolism) clearly indicated the enhancement of several specific pathways. The most represented among these were the phenylpropanoid pathway (with four enzymes—phenylalanine ammonia-lyase, cinnamyl alcohol dehydrogenase-like protein and two caffeic acid *O*-methyltransferases) and the flavonoid biosynthesis pathway (with four enzymes—2-hydroxyisoflavanone dehydratase, 2-hydroxyisoflavanone dehydratase and two chalcone-flavonone isomerase family proteins). In addition, phospho-2-dehydro-3-deoxyheptonate aldolase (the first enzyme in the chorismate biosynthesis pathway) and cytochrome P450 family monooxygenase G7KQK8 (the enzyme involved in the biosynthesis of xantophylls) were identified as up-regulated in the presence of the source of mineral nitrogen. Further, enhanced expression of the FAD-binding berberine family protein (an oxidase involved in biosynthesis of benzylisoquinoline alkaloids) was associated with these soil conditions. Finally, dirigent protein, the enzyme involved in the coupling of phenoxyl radicals, which is associated with all these pathways, was found to be more abundant when mineral nitrogen-containing salts were added to the soil. The enzyme families (bin 26) were represented by secretory proteases (plant basic secretory protein (BSP) family protein), stress tolerance (carboxylesterase), oxidoreductase of furanone biosynthesis (2-methylene-furan-3-one reductase) and others. Finally, the up-regulated redox-related pathways were represented by peroxidases, glutathione reductases and oxidoreductases.

The down-regulation pattern of the k-3358 root proteome was dominated by protein metabolism (bin 29), secondary (bin 16) and redox metabolism (bin 21, Figure 3a, Appendix A). The first group appeared to be heterogeneous: along with three polypeptides involved in proteasomal degradation and one S10 family carboxypeptidase, several players of the translation machinery (elongation factor Tu and SRP9/SRP14 subunit of the signal recognition particle), protein translocation (three different signal peptidases) and folding (peptide methionine sulfoxide reductase and protein disulfide-isomerase) machinery were discovered. The nitrogen assimilation bin (six accessions) represented the most strongly down-regulated group. This alteration could explain the strong enhancement of the protein degradation machinery described in the previous paragraph. The five down-regulated enzymes of bin 16 represent five diverse pathways of secondary metabolism: the phenylpropanoid pathway (caffeic acid/5-hydroxyferulic acid *O*-methyltransferase), metabolism of oximes (cytochrome P450 83B1) that might contribute to metabolism of auxins [22], alkaloid biosynthesis (3′-hydroxy-*N*-methyl-(S)-coclaurine 4′-*O*-methyltransferase), detoxification of α-dicarbonyls in the glyoxalase cycle (lactoylglutathione lyase-like protein) and oxidative polymerization of phenolics (dirigent protein). Moreover, aspartate and cysteine metabolism were clearly suppressed (Appendix A).

The pattern of the differential proteome response of the k-8274 roots to external mineral nitrogen supplementation (Appendix A) differed from that of the k-3358 roots. Thus, redox metabolism (bin 21), miscellaneous enzyme families (bin 26), protein metabolism (bin 29) and secondary metabolism (16) appeared to be the most responsive to these conditions (Figure 3b, Appendix A). Among them, bin 21 was the best represented and accounted for 11 entries, being similar to the response of k-3358 in both protein numbers and diversity. Bin 26 was represented by six entries, which included an oxidoreductase, a glycosyl hydrolase, α/β-hydrolase and carboxypeptidase (Appendix A). In contrast to the roots of the k-3358 genotype, no up-regulation of the proteasome protein degradation machinery was observed. On the other hand, peptidases, potentially involved in lysosomal degradation (family C1 peptidase) and stress-induced programmed cell death (metacaspase-4-like protein) were observed as up-regulated, as well as the proteins involved in protein biosynthesis and maturation (aspartate-tRNA ligase 2, cytoplasmic-like, acyl-peptide hydrolase-like protein). Finally, two unique proteins (phenylcoumaran benzylic ether reductase-like protein Fi1 and vestitone reductase-like protein), along with three of those detected in k-3358, could be found among the five up-regulated proteins representing bin 16 (secondary metabolism).

The down-regulation pattern of the k-8274 root proteome was less abundant than the up-regulation one and dominated by the polypeptides involved in protein metabolism (bin 29), nitrogen assimilation (bin 12) and signaling (bin 30, Figure 3b and Appendix A). The down-regulated part of bin 29 was more abundant and diverse in comparison to the up-regulated one. Thus, among the ten accessions, three proteins were involved in protein degradation (one proteasome protein, one associated regulatory protein and one peptidase S10 type), whereas five proteins appeared to be involved in protein synthesis and translocation and two in protein folding (Appendix A). The proteins involved in nitrogen metabolism were represented by six leghemoglobin accessions, which were the most strongly down-regulated species (although at least three times less affected in comparison to the root proteome of k-3358). Finally, DNA-binding regulatory element (general regulatory factor 2 of the 14-3-3 family) was detected among the four down-regulated signaling proteins. Moreover, similarly to k-3358, cysteine and aspartate metabolism were affected in the roots of the k-8274 grown in the presence of mineral nitrate, although this response was much less metabolically diverse in comparison to k-3358. On the other hand, in contrast to k-3358, here, threonine metabolism was affected as well.

The nitrogen assimilation bin (six accessions) represented the most strongly down-regulated group. This alteration could explain the strong enhancement of the protein degradation machinery described in the previous paragraph. The five down-regulated enzymes of bin 16 represent five diverse pathways of secondary metabolism: phenylpropanoid pathway (caffeic acid/5-hydroxyferulic acid *O*-methyltransferase), metabolism of oximes (cytochrome P450 83B1) that might contribute to metabolism of auxins [22], alkaloid biosynthesis (3′-hydroxy-*N*-methyl-(S)-coclaurine 4′-*O*-methyltransferase), detoxification of α-dicarbonyls in the glyoxalase cycle (lactoylglutathione lyase-like protein) and oxidative polymerization of phenolics (dirigent protein). Moreover, aspartate and cysteine metabolism were clearly suppressed (Appendix A).

When the k-3358 plants were grown in the absence of mineral nitrogen sources, but in the presence of legume–rhizobial symbiosis, the response of the root proteome was minimal and covered nine up- and five down-regulated proteins (Figure 3c and Appendix A). Thus, bin 29 (protein metabolism) showed the most pronounced up- and down-regulation response (three accessions in each case). All three up-regulated proteins represented the biosynthetic branch of the protein metabolism—two ribosomal proteins and one storage polypeptide, glutelin type-B-like protein. In contrast, the down-regulation pattern of this bin was heterogeneous—one catabolic protein (carboxypeptidase of the S10 family), one repairing enzyme (F-box plant-like protein) and one regulatory polypeptide (peptide methionine sulfoxide reductase-like protein). The latter pattern was accompanied by one leghemoglobin—a protein involved in nitrogen fixation, which decreased its abundance. The up-regulation pattern included enzymes involved in secondary metabolism (phenyl-alanine ammonia-lyase and dirigent protein), transport (transmembrane amino acid transporter family protein and drug resistance transporter-like ABC domain protein) and signaling (B-5-like nuclear transcription factor and vacuolar-sorting receptor-like protein).

In agreement with our earlier observations at the level of seeds [18], the proteome responses were essentially enhanced when the roots of the k-3358 and k-8274 plants were inoculated with rhizobial culture in parallel with the infection with arbuscular mycorrhiza fungi (AMF+NB, Figure 3d,e, Appendix A). Thus, functional annotation of the k-3358 root proteome, which appeared to be highly responsive to such combined inoculation, revealed 10 (16 accessions) and 12 bins (17 accessions) representing up-regulation and down-regulation patterns, respectively (Figure 3d and Appendix A). Among the former, protein metabolism (bin 29) and miscellaneous enzyme families (bin 26) were the most represented, with four and three accessions, respectively (Appendix A). The polypeptides representing the bin “protein metabolism” were mostly involved in protein biosynthesis (leucine tRNA ligase, cyto-plasmic isoform X1) and folding (TCP-1/cpn60 chaperonin family protein), while the annotated enzymes (bin 26) were involved in phenolic metabolism—caffeic acid *O*-methyltransferase (phenylpropanoid pathway), 2-hydroxyisoflavanone dehydratase (flavonoid biosynthesis pathway) and putative secoisolariciresinol dehydrogenase (lignin biosynthesis). Although the down-regulated proteins were most represented in terms of bins, only three annotated functional classes could be observed with more than one differentially abundant protein. The first of these, bin 29, included five accessions, three of which were catabolic proteins (26S proteasome regulatory particle non-ATPase subunit 8 and two S10 family peptidases). The other two down-regulated proteins of this bin represented regulatory (plant F-box-like protein) and folding (70 kDa heat shock protein chaperone DnaK) functions. Bins 30 (signaling) and 11 (lipid metabolism) were represented with two proteins each. Thus, two signaling proteins (calmodulin and putative serine/threonine-protein kinase) and two enzymes of lipid metabolism (fatty acyl-CoA synthetase family protein and phospholipase A1-IIgamma-like isoform X1) were annotated to these two protein functional classes, respectively.

In comparison to k-3358, the k-8274 plants demonstrated approximately 2.5–3.5-fold more abundant response to combined inoculation with two microorganisms, which featured also different bin distribution. Indeed, the up-regulation pattern included 39 proteins distributed in 22 functional classes, while the down-regulation pattern had 59 accessions in 26 bins (Figure 3e and Appendix A). Redox metabolism comprised the most representative up-regulated functional class. All seven proteins contributing to this bin appeared to be protein antioxidants and ROS-detoxification agents: ascorbate oxidase, monodehydroascorbate reductase, glutathione S-transferase and several peroxidases (Appendix A). The further four bins, namely protein metabolism, miscellaneous enzyme families, stress response and mitochondrial energy metabolism, were represented by three up-regulated proteins each. The latter group included three FAD- and NAD(P)H-dependent oxidoreductases (quinine reductases) of the mitochondrial electron-transport chain. The stress response proteins included one late embryogenesis abundant (LEA) protein and one heat shock protein that might indicate (in addition to the above-listed redox proteins) activation of several mechanisms of stress response. Importantly, one of these proteins, PLAT-plant stress protein, plays an important role in successful nodule formation [23]. The enzyme families were represented with one *O*-methyltransferase, glycoside hydrolase 18 domain-containing protein (with chitinase and/or ENGase activity) and 3-ketoacyl-CoA thiolase peroxisomal-like protein, which is known as an important factor in fatty acid peroxisomal beta-oxidation cleaving straight chain 3-oxoacyl-CoAs of different length. Finally, bin 29 (protein metabolism) was represented by three proteins, including one heat shock protein and Kunitz type trypsin inhibitor.

The down-regulation pattern was more abundant and diverse (Figure 3e and Appendix A). Although it was dominated with bin 29 (protein metabolism), proteins involved in stress response, amino acid and nitrogen metabolism demonstrated a strong abundance decrease as well. The down-regulated polypeptides (eight accessions) of bin 29 were mostly represented by the anabolic branch of the protein metabolism, i.e., proteins involved in translation (elongation factor Tu and an S40 ribosomal protein), translocation (two signal peptidases) and folding (one mitochondrial heat shock 70 kDa protein and one peptide methionine sulfoxide reductase-like protein), with only minimal impact of the catabolic pathways (proteasome subunit alpha type; Appendix A). The next most abundant down-regulated protein group was bin 20 (stress), which was represented by two heat shock proteins, early nodulin ENOD18 (which is known as a stress-responsive protein, which is over-expressed e.g., under drought conditions [24]), chloroplastic plastoglobulin-1 (which is involved in the transport of lipids across the tilacoid membrane [25]) and carboxylesterase 15, an enzyme involved in catabolism of strigolactones [26]. A decrease in the abundance of the latter enzyme (which is known to degrade strigolactones [26]) might indicate the success of the arbuscular mycorrhiza symbiosis, as strigolactones are directly involved in its establishment. Interestingly, the pathways of arginine, aspartate and cysteine biosynthesis appeared to be suppressed in the presence of the combined root inoculation with two microorganisms. Analogously, four leghemoglobins were strongly down-regulated under these conditions.

### 2.5. Subcellular Localization

The prediction of subcellular localization was performed using LocTree3, with subsequent manual verification based on the UniProt database and literature data (Appendix A). The results demonstrated that the whole cell responded due to strong metabolic interconnection of the organelles. There were rather similar patterns of differentially abundant proteins, changing their localization in all experimental groups of plants grown in the presence of different soil supplementations. For both genotypes, cytosol, plastids and extracellular space appeared to be the most strongly contributing compartments (Figure 4a–e). On the other hand, although it could be noted that proteins localized in the nucleus and Golgi apparatus were also highly susceptible to changes in the roots of the k-3358 plants, this was much less the case for the genotype k-8274 (Figure 4b).

## 3. Discussion

The principal objective of this work was to study the root response of two pea (*P. sativum*) genotypes, namely k-3358 and k-8274, to inoculation with beneficial soil microorganisms. Based on the dynamics of multiple productivity parameters (including morphological traits [27] and seed proteome shifts [18]) related to associations with beneficial soil microorganisms (BSMs), these genotypes are known to be low- and high-performing, respectively. Indeed, in a three-year field trial, k-3358 did not increase its seed productivity due to inoculation with BSMs, whereas, for k-8274, the increment of seed weight due to inoculation exceeded 100% [28]. The genetic basis of these differences in the symbiotic responsivity of garden pea has been previously investigated, and some related genes were identified [12]. In particular, in a vegetation experiment accomplished in non-sterile soil, three unrelated pea genotypes with high EIBSM were compared with three low-EIBSM genotypes under double inoculation with rhizobia and AM fungi. Analysis of the root transcriptome gave access to a list of differentially expressed genes, with higher transcript levels observed in the roots of three highly responsive genotypes, as compared to the roots of three low responsive ones [29]. This list includes several genes related to flavonoid biosynthesis and oxidative stress response, which may suggest that the molecular bases of high EIBSM is linked to higher flavonoid production and maintenance of the redox homeostasis [29].

### 3.1. Inter-Genotype Differences in Proteome Signatures

Here, to get access to the molecular bases of high EIBSM trait, we address the changes in the root proteome associated with inoculation of the pea k-3358 and k-8274 plants with rhizobia in the absence (nodule bacteria, NB) and presence of simultaneous infection with arbuscular mycorrhiza fungi (AMF+NB), in parallel to the plants grown in the absence (controls) and presence of mineral nitrogen supplementation (mineral nutrition, MN). The phenotypes of the corresponding plants, as well as the details of the experimental setup, were comprehensively described by Zhukov et al [27]. It was shown that soil supplementation with nitrate resulted in pronounced decreases in the numbers of nodules formed on the roots of both k-8274 and k-3358 plants (in comparison to the controls grown without mineral and/or microorganism soil supplementations). The decrease in the numbers of nodules, was also observed for the high-EIBSM k-8274 genotype (but not for the low-EIBSM k-3358 one) in the presence of rhizobia and AM fungi (although the mycorrhization parameters were not addressed in that work) [27].

We believe that this experimental setup might give deeper insight in the molecular mechanisms behind the plant response to interaction with both microorganisms. Both the differences in the effect of the legume–rhizobial symbiosis on the root proteome and the impact of AM fungi on its dynamics could, therefore, be efficiently addressed. Obviously, this will result in pronounced success in the establishment of fundamentally new pea cultivars that are capable of efficient interaction with beneficial soil microorganisms.

The first step to understand the responsivity of two genotypes to symbiosis with BSMs was their comparison in the absence of microbial inoculation and supplementation of mineral nitrogen sources. Although this comparison revealed relatively low numbers of differentially expressed proteins (that could indicate minimal inter-genotype differences visible in the absence of symbiosis; Appendix A), their functional annotation could give access to some genotype-specific features (Appendix A).

Thus, in the absence of the mineral nitrogen supplementation, both genotypes are expected to suffer from nutritional stress. However, this state appeared to be much more pronounced in the roots of the k-8274 plants. Indeed, the latter showed essentially higher diversity of the stress-induced polypeptides in comparison to the k-3358 plants (seven vs. two accessions), with strong domination of redox proteins. This response pattern was in agreement with higher expression of the proteins involved in the biosynthesis of phenylpropanoids and their polymerization in the cell wall, that was clearly characteristic for the k-8274 roots. This reaction is characteristic for the abiotic stress response [30].

On the contrary, no substantial activation of the redox metabolism (and, specifically, no up-regulation of peroxidases) could be seen in the roots of the k-3358 plants, while the only two stress-associated proteins dominating here were related to abiotic stress tolerance [31,32]. In agreement with this, MYB-CC type transcription factor, which is known to be involved in regulation of the cell cycle, cell morphogenesis, responses to biotic and abiotic stresses [33], was found to be most expressed in the k-3358 roots (Appendix A). Analogously, cysteine peptidase C1 and serine carboxypeptidase S10, which are involved in the regulation of plant development, seed maturation, senescence and stress response [20,34], were more abundant in the k-3358 roots. Most likely, this protein response pattern indicates higher stress tolerance and ability to sustain cell integrity of this genotype.

### 3.2. Subcellular Root Proteome Response of the k-3358 and k-8274 Plants to Soil Complementation with Mineral Nitrogen Localization

The inter-genotype differences became more striking when soil was supplemented with mineral nitrogen-containing salts. The analyzed genotypes demonstrated differences in their sensitivity to this factor, with the genotype k-3358 being essentially more responsive. The main feature was an essential decrease in the expression of leghemoglobins and several other regulatory proteins and enzymes related to nitrogen fixation. Such suppression of symbiotic nitrogen fixation in the presence of mineral nitrogen in soil has been reported for bean [35], soybean [36] and pea [37] (in the experiment from which the material for this study was collected, the suppression of nodulation under mineral nutrition was also noted for both k-3358 and k-8274 [27]). However, the relative abundance of leghemoglobins demonstrated at least a 7-fold stronger decrease in the presence of mineral nitrogen in the roots of k-3358, in comparison to the effect of the same conditions in the k-8274 plants (Appendix A). Interestingly, the roots of k-3358 showed the strongest suppression of expression of the heat shock protein DnaJ (1391-fold). To some extent, this can be explained by its role in the improvement of symbiosis performance (as reviewed by Da Silva and co-workers [38]).

In general, the addition of inorganic nitrogen to the soil resulted in strong down-regulation of multiple proteins, critically impacting the success and efficiency of legume–rhizobial symbiosis. Thus, among the proteins strongly down-regulated in the roots of k-3358 in the presence of mineral nitrogen were the proteins involved in initiation of symbiosis—early nodulins 7 (which is typically expressed in pea root nodules induced by *Rhizobium leguminosarum* bv. *viciae* [39]) and 18 (which can act as a carbohydrate transporter [40] or might impact on salt stress tolerance [41]). Following this logic, a pronounced decrease in the relative abundances of VWA domain containing CoxE-like protein, putative chromatin regulator PHD family A0A396JLK8 (both 89-fold), FAM91A1-like protein (10-fold) and pentatricopeptide repeat-containing protein At1g31920 (53-fold) might indicate regulation of nodulation.

Another strongly down-regulated protein was Acyl-CoA-binding protein, ACBP (75.5-fold). This protein is a key player in oxygen sensing in plants (which relies on the N-end rule pathway, NERP), it binds group VII ethylene response factor and, thus, is directly involved in the regulation of their stability [42]. A slightly lower but pronounced (49-fold) nitrogen-induced decrease in the abundance of the vegetative lectin Q41069, the molecule known to impact the success of root inoculation with rhizobia [43], was observed as well. Finally, several enzymes that are critical for symbiosis establishment, were also strongly down-regulated. This was the most pronounced for carbonic anhydrase (129-fold). The gene of this enzyme is known to be induced in the nodules of different origin, and the activity of this enzyme might have a clear anabolic role [44]. Another important actor, cell wall invertase, which degrades sucrose to make sugars accessible for the symbiotic microorganisms [45], was also strongly down-regulated in k-3358 roots upon N supplementation (89-fold). Interestingly, although multiple enzymes of central metabolism were down-regulated as well (e.g., fructose-bisphosphate aldolase, isocitrate dehydrogenase and cytochrom c), the degree of this suppression of expression was relatively low. Finally, clear down-regulation of heme biosynthesis (oxygen-dependent coproporphyrinogen-III oxidase), ammonia metabolism (amine oxidase and aspartate aminotransferase) and cysteine metabolism (cysteine synthase and lactoylglutathione lyase-like protein) was observed (Appendix A).

Obviously, the described down-regulation pattern observed in the k-3358 roots was the result of the strong enhancement in biosynthesis of enzymes and regulators primarily associated with protein degradation (amino- and carboxypeptidases, proteinases, components of proteasomes; Appendix A). This was, however, most likely accompanied by enhanced transcription. Indeed, two up-regulated proteins act at the chromatin level: the chromatin CoxE-like regulator PHD family (PHD-containing proteins also mediate adaptive responses of the root to environmental factors—a part of the chromatin remodeling system, mediating activation of gene expression [46]) and histone H2B, one of the main components of the nucleosome. Importantly, the synthesis of core histones is strictly coupled to DNA replication in the S-phase. This pattern of up- and down-regulated proteins may indicate the intensification of processes related to maturation.

The comparison of the above-described responses with those of the k-8274 plants showed that the whole pattern of the nitrogen-responsive proteins was less represented among the differentially expressed root proteome and the observed changes were less pronounced, although in further details similar (Appendix A). Thus, the genotypes showed similar degrees of stress response—both in terms of the types and numbers of redox enzymes and in terms of fold change (although k-3358 showed a more abundant up-regulation pattern of stress-induced proteins).

However, despite this similarity, in contrast to the genotype k-3358, the k-8274 plants did not show any activation of protein degradation in response to supplementation with mineral nitrogen (although down-regulation of three proteins involved in this process, including one proteasome component, was observed). Another important difference was the presence of the N-induced responses at the level of secondary metabolism. Thus, although both genotypes showed up-regulation of phenolic metabolism (specifically, phenylpropanoid pathway and flavonoid biosynthesis), it was essentially more pronounced for k-3358 (Appendix A). In line with this, dirigent protein, which plays an important role in lignan and lignin biosynthesis by exercising the spatial control of lignin deposition, especially under stressed conditions [47], was more strongly up-regulated in the k-3358 roots (2.5- vs. 1.5-fold). In contrast, the k-8274 plants showed a more pronounced decrease in the abundance of lactoylglutathione lyase, which might indicate less activity of glyoxalase cycle and, hence, higher amenability of this genotype to oxidative stress under these conditions.

Thus, it can be concluded that the k-3358 plants appeared much more responsive to supplementation with mineral nitrogen salts, demonstrating higher adaptation potential with this factor. It showed a more pronounced redirection of metabolism from symbiotic nitrogen nutrition to the mineral nitrogen supply. Thus, this genotype had higher (in comparison to k-8274) efficiency when nitrogen was supplied from mineral sources. In contrast, k-8274 showed compromised metabolic efficiency under such nutritional conditions, which was in line with its lower productivity gain when mineral nitrogen sources were used.

### 3.3. Root Proteome Response of the k-3358 and k-8274 Plants to Inoculation with Symbiotic Rhizobia

When the nitrogen supply was provided only via the symbiotic route (i.e., upon establishment of legume–rhizobial symbiosis in the absence of soil supplementation with mineral nitrogen sources), no changes in the proteome of the k-8274 roots could be observed. The absence of detectable responses at the levels of metabolic pathway, regulatory systems and stress response might indicate good compatibility of the plant and microbial partners. This fact was in good agreement with the good responsivity of this genotype to inoculation with *R. leguminosarum* [18].

In contrast, the roots of the k-3358 plants showed a detectable response at the proteome level, although it was minimal (Appendix A). The up-regulation pattern was represented with a small number of bins, with no more than one to two proteins in each. We can pay attention to the increase in glutelin type-B-like protein, one of the main reserve proteins that are combined into protein bodies in the endosperm, which may indicate an acceleration of plant development. This fact is in a good agreement with the seed productivity data [18], indicating low responsivity of this genotype to inoculation with symbiotic rhizobia and lower biomass gain due to accelerated development.

### 3.4. Root Proteome Response of the k-3358 and k-8274 Plants to Combined Inoculation with Nodule Bacteria and AM Fungi

As the last step, we addressed the response of the root proteomes of the k-3358 and k-8274 pea genotypes to combined inoculation with arbuscular mycorrhiza fungi (AMF) and rhizobia. Based on the previously acquired and published data [18], this setup results in maximal productivity gain of the genotype k-8274, in which responsivity to rhizobial inoculation is increased when the roots are colonized with AM fungi.

Our results obtained in this study indicated that both genotypes demonstrated clear proteome responses, although they were more pronounced for the k-8274 plants. Indeed, the latter showed approximately two times more pronounced up- and three times higher down-regulation response (Appendix A). Also, the response of the k-8274 plants was featured with much higher functional diversity—22 and 26 bins were up- and down-regulated, respectively, while only 10 bins contributed to the up-regulation pattern in the k-3358 roots. This fact reflects the higher responsivity of k-8274 to inoculation with beneficial microorganisms, as compared to k-3358.

In the roots of k-3358, histone H2B.1 (a variant form of histone H2B) was detected. Also, polypeptides involved in protein synthesis and maturation (leucine tRNA ligase, TCP-1/cpn60 chaperonin family protein) were up-regulated. On the other hand, in the k-8274 plants this functional group was represented by only three proteins—heat shock protein with ATPase domain, Kunitz type trypsin inhibitor and uncharacterized protein.

In the k-3358 roots, enzymes of phenylpropanoid pathway, flavonoid biosynthesis pathway and lignin biosynthesis were clearly up-regulated. This may indicate the activation of defense reactions as a response to mycorrhization. Putative 3-oxoacyl-[acyl-carrier-protein] reductase, an enzyme involved in fatty acid biosynthesis (a process accompanying the arbuscular mycorrhiza development in plant roots) [48], was also found to be up-regulated. Down-regulated proteins belong to almost all functional groups, but most of the groups are represented by a single protein. Bin 29 (Protein metabolism), represented by Carboxypeptidases, 26S proteasome regulatory particle non-ATPase subunit 8, HSP DNA K and F-box plant-like regulatory protein, deserves attention. Then, a down-regulation of isocitrate dehydrogenase [NADP], an enzyme participating in the antioxidant system [49], and 1-aminocyclopropane-1-carboxylate oxidase, an enzyme catalyzing the last step of ethylene biosynthesis, was noted. This may indicate that biological processes in k-3358 roots are less active under NB+AMF inoculation than in control conditions.

For the genotype k-8274, the greatest changes affected the group of proteins of redox metabolism: protein antioxidants and ROS-detoxification agents, which contribute to the overall activation of metabolism and increased adaptive capacity of the plant. It is known that establishment of mycorrhiza is accompanied by activation of plant immune responses (both local and systemic). This phenomenon is generally referred to as priming of plant immunity, or (in the context of this plant–microbial interaction), mycorrhiza-induced resistance [50]. This activation leads to a primed state of the plant with pre-activated defense mechanisms, which include activation of ROS-dependent signaling and accumulation of phenolic compounds [51]. Our data suggest that k-8274 plants could be primed by mycorrhization, whereas this mechanism was not activated in the k-3358 plants. The up-regulation of the stress-protective proteins, like the late embryogenesis abundant (LEA) family, PLAT-plant-stress protein and lipoxygenase, might lead to the same effect, i.e., to the increase of adaptive capacity of plants. Also, a number of up-regulated functional classes of proteins were represented by one or two proteins. The related shifts indicated enhancement of catabolism, primarily carbohydrate catabolism: glucan endo-1,3-beta-glucosidase (polysaccharide cleavage), fructose-bisphosphate aldolase (glycolysis), isocitrate dehydrogenase and malic enzyme (tricarboxylic acid cycle). The up-regulation of components of the respiratory complexes of the mitochondrial inner membrane and ferredoxin-NADP reductase of chloroplasts is also consistent with these data. Arginase appeared to be one of the enzymes, which were up-regulated in the plants inoculated with AMF and rhizobia. Arginase is involved in mobilization of availablenitrogen by arginine accumulation, which contributes to the synthesis of other amino acids and polyamines.

The down-regulation pattern turned to be even more abundant in terms of both diversity and relative abundance shifts observed for individual proteins. Similar to the proteome of the roots of the former genotype, the most representative is the functional group “protein metabolism”, represented by factors of translation, folding and processing. Also, a number of stress-induced proteins represented by two heat shock proteins, early nodulin ENOD18 and leghemoglobins, together with proteins from the pathways of arginine, aspartate and cysteine biosynthesis, were down-regulated. The observed down-regulation of nodule-specific proteins was in a good agreement with the fact that the numbers of nodules on the roots of the k-8274 plants upon inoculation with NB+AMF inoculation were lower in comparison to control roots [27]. It might indicate that control of the nodule numbers might contribute to the manifestation of the high EIBSM trait in the k-8274 genotype.

The down-regulation of the polypeptides associated with nodule development in the roots of the k-8274 plants inoculated with nodule bacteria and arbuscular mycorrhizal fungi may be a result of the plant defense priming. The degree of the priming effect due to inoculation with microorganisms may be different in different accessions within a plant species [52,53]. This seems to be a possible mechanism behind the manifestation of the EIBSM trait under natural conditions, when not only beneficial but also deleterious interactions (e.g., with pathogenic microorganisms present in soil) are possible.

## 4. Materials and Methods

### 4.1. Reagents

Unless stated otherwise, the following materials were obtained from Merck KGaA (Darmstadt, Germany): ethylenediaminetetraacetic acid, phenol, dithiothreitol, iodoacetamide, Coomassie Brilliant Blue G-250; SERVA Electrophoresis GmbH (Heidelberg, Germany): β-mercaptoethanol; AppliChem (Darmstadt, Germany): phenylmethylsulfonyl fluoride; Hollywell (Charlotte, NC, USA): formic acid (LC-MS Grade), acetonitrile (LC-MS Grade); Progenta™ (Radnor, PA, USA): Anionic Acid Labile Surfactant II (AALS II). All other chemicals were purchased from Carl Roth GmbH and Co (Karlsruhe, Germany). Water was purified in-house in a water conditioning and purification system, the Millipore Milli-Q Gradient A10 system (resistance 18 mΩ/cm, Merck Millipore, Darmstadt, Germany).

### 4.2. Plant Experiment

In this study, we used plant material from the previously completed vegetation experiment described in detail earlier [27]. Pea plants of the genotypes k-8274 (cv. Vendevil, originated from Vilmorin, La Ménitré, France, and propagated in the collection of cultivated peas, N. I. Vavilov All-Russian Institute of Plant Genetic Resources (VIR) St. Petersburg, Russia and k-3358 (unnamed cultivar from Saratov region, Russia, from the collection of cultivated peas, N. I. Vavilov All-Russian Institute of Plant Genetic Resources (VIR) St. Petersburg, Russia), were grown in non-sterile soil (three plants per five-liter pot) in a vegetation house during the summer period at ARRIAM, St. Petersburg, Russia. The experiment included the following variants: non-inoculated plants (control); plants treated with full mineral nutrition (MN)—a mineral nitrogen reference group (supplemented with ammonia nitrate); plants inoculated with *R. leguminosarum* bv. viciae strain RCAM 1026 (NB); plants simultaneously inoculated with *R. leguminosarum* bv. viciae strain RCAM 1026 and with a mixture of arbuscular mycorrhizal fungi *Rhizophagus irregularis* BEG144, *R. irregularis* BEG53 and *Glomus* sp. ST3 (NB+AMF). After four weeks of growth, the plants were harvested; root systems were rinsed with water, immediately frozen in liquid nitrogen and stored at −80 °C until further analysis. All experiments were performed in three biological replicates. Thus, three plants in one pot were pooled per replicate.

### 4.3. Protein Extraction

Protein extraction was performed according to the protocol of Frolov and co-workers, with minor modifications [54]. In detail, 650 µL of ice cold (4 °C) phenol extraction buffer (0.7 mol/L sucrose, 0.1 mol/L KCl, 5 mmol/L ethylenediaminetetraacetic acid (EDTA), 2% (*v*/*v*) β-mercaptoethanol and 1 mmol/L phenylmethylsulfonyl fluoride (PMSF) in 0.5 mol/L Tris-HCl buffer, pH 7.5) was added to approximately 500 mg of ground frozen root material. After a short vortexing (30 s), 650 µL of ice cold phenol (4 °C) pre-saturated with 0.5 mol/L Tris-HCl buffer (pH 7.5) was added and vortexed under the same conditions. Afterwards, the samples were shaken (30 min, 900 rpm, 4 °C) and centrifuged (5000× *g*, 15 min, 4 °C), the lower aqueous phase was discarded, and the phenolic phase was washed two times with equal volumes of the phenol extraction buffer. Each buffer addition was followed by vortexing for 30 s, shaking (30 min, 900 rpm, 4 °C) and centrifugation (5000× *g*, 15 min, 4 °C). Finally, the aqueous phase was discarded, the phenol phase was supplemented with five volumes of cold (−20 °C) ammonium acetate in methanol (0.1 mol/L), and the proteins were left to precipitate at −20 °C overnight. The next morning, the total protein fraction was pelleted by centrifugation (5000× *g*, 15 min, 4 °C), and the pellets were washed twice with two volumes (relative to the volume of the phenol phase before precipitation) of cold (4 °C) methanol, and once with the same volume of cold (4 °C) acetone. Each time, after re-suspending, the samples were centrifuged (5000× *g*, 10 min, 4 °C). Finally, the cleaned pellets were dried under air flow in a fume hood for 1 h and then re-constituted in the shotgun buffer (8 mol/L urea, 2 mol/L thiourea in 0.1 mol/L Tris-HCl buffer, pH 7.5) containing 0.1% (*w*/*v*) anionic acid-labile surfactant II (AALS II, (Protea, USA).

### 4.4. Determination of Protein Concentration

Determination of protein concentration was performed using a 2-D Quant Kit (GE Healthcare, Chicago, IL, USA), according to the manufacturer’s protocol. The precision of the assay was cross-verified by SDS-PAGE, according to Greifenhagen et al., with some modifications [55]. In brief, after the gels were stained overnight with 0.1% (*w*/*v*) Coomassie Brilliant Blue G-250 for 12 h, the average densities across individual lanes (expressed in arbitrary units) were determined by the Bio-5000plus Microtek imaging system controlled by BioDoc analysis software (v.2.66.3.22) (SERVA Electrophoresis, Heidelberg, Germany). For calculation of the relative standard deviations (RSDs), the densities of individual lines were normalized to the gel average value by ImageJ software (v. 1.54a).

### 4.5. Tryptic Digestion

The aliquots of the tryptic digests (50 µg) were supplemented with the AALS II-containing shotgun buffer to obtain the final sample volumes of 90 µL. Afterwards, 10 µL of 50 mmol/L tris(2-carboxyethyl)phosphine) in AALS-free shotgun buffer was added. After a 30-min incubation at 37 °C under continuous shaking (450 rpm), 11 µL of 100 mmol/L iodoacetamide in the same buffer was added and the samples were further incubated (1h, 450 rpm, 4 °C) before 875 µL of aqueous 50 mmol/L ammonium bicarbonate (ABC) and freshly prepared 0.5 µg/µL trypsin in ABC (enzyme-to protein-ratio of 1:20 *w*/*w*) were added. After incubation for 5 h (450 rpm, 37 °C), digestion was repeated at the enzyme-to protein-ratio of 1:50 (*w*/*w*) overnight. The next morning, the digests were frozen and the samples were stored at −20 °C. The completeness of the digestion was verified by SDS-PAGE, as described by Greifenhagen and co-workers [55].

### 4.6. Solid Phase Extraction

After decomposition of ALLS (10% *v/v* TFA, 20 min at 37 °C), the proteolytic hydrolysates were pre-cleaned by reversed phase solid phase extraction (RP-SPE), using the elution scheme of Spiller et al. [56], with minor modifications. Stage tips with six layers of C-18 Extraction Disks (66883-U) (Sigma-Aldrich, Burlington, MA, USA) were prepared in 200 µL polypropylene pipette tips and inserted into 2 mL tubes via plastic tube adaptors. The stage tips were conditioned with 100 µL of MeOH (2000× *g*, 5 min, 25 °C) and equilibrated with two portions of 200 µL of 0.1% (*v*/*v*) formic acid (FA, 2000× *g*, 5 min, 25 °C), before the individual tryptic digests were applied and the stage tips were centrifuged (2000× *g*, 5 min, 25 °C). After washing (2 × 200 µL of 0.1% (*v*/*v*) FA) with centrifugation after each step (2000× *g*, 5 min, 25 °C), the peptides were eluted by sequential application of 150 µL of 60% (*v*/*v*) acetonitrile and 150 µL of 80% (*v*/*v*) acetonitrile, 0.1% (*v*/*v*) FA. Finally, the pooled eluate was transferred to 0.5 mL polypropylene tubes and dried (4 °C) under reduced pressure in a Concentrator Plus vacuum concentrator (Eppendorf, Germany).

### 4.7. Nano LC-MS/MS

The protein hydrolysates were loaded on a PepMap 100 C18 trap column (300 μm ×5 mm, 3 μm particle size) and separated on a PepMap RSLC C18 column (75 μm ×250 mm, 2 μm particle size) separation column (both Thermo Fisher Scientific, Bremen, Germany) using a nLC 1000 nanoflow chromatography system (Thermo Fisher Scientific, Bremen, Germany) equipped with an EASY-Spray™ Source and coupled online to a LTQ-Orbitrap XL hybrid mass spectrometer (Thermo Fisher Scientific, Bremen, Germany). The details of the chromatographic separation method and MS-settings, including the applied data-dependent acquisition (DDA) parameters, are summarized in Appendix A. The acquired raw data are available in the PRIDE repository under the project accession number PXD058701 and project DOI 10.6019/PXD058701.

### 4.8. Data Processing and Post-Processing

Identification of peptides and annotation of proteins was performed on a combined sequence database, designed FASTA files of *Lotus japonicus*, *Medicago truncatula* and *Pisum sativum* from UniProtKB, combined into a single file (*Pisum sativum* V1r port) as described by Mamontova et al [19]. The duplicated sequences were recognized and deleted using the Cd-hit program. The database search relied on the SEQUEST engine, operated in the framework of Proteome Discoverer v. 2.2 software (Thermo Fisher Scientific, Bremen, Germany). For the search settings, see Appendix A. Quantitative protein analysis (label-free quantification) relied on the Progenesis QI software (v.4.2) with the FDR correction at the confidence level *p* ≤ 0.05. Further, Mercator4 (v. 2) was used to perform functional annotation of differentially expressed proteins [57,58]. Prediction of sub-cellular localization of those proteins was performed using the LocTree3 tool. The annotation and prediction data were manually verified to curate possible misidentifications.

## 5. Conclusions

The EIBSM (effectiveness of interaction with beneficial soil microorganisms) trait in the pea is manifested by an increase in seed weights when the plants are grown in symbiosis with nitrogen-fixing rhizobia and arbuscular mycorrhiza fungi. As we showed before, this effect can be (at least partly) explained by prolongation of the seed filling period. Accordingly, the phenotypic differences between the high- and low-EIBSM genotypes could be underlied by different ecological strategies. Specifically, the high-EIBSM k-8274 plants are characterized by a prolonged seed filling period and relatively low numbers of formed seeds (the K-strategy), whereas the low-EIBSM k-3358 plants are characterized by accelerated seed maturation with high quantities of formed seeds (the r-strategy) [18]. Each strategy can be more or less beneficial, depending on the environment. Here, we addressed the features of pea root proteome, which accompany the EIBSM trait. Using the proteomics approach, we found that the high-EIBSM genotype k-8274 demonstrated a more pronounced response to inoculation with NB+AMF than the low-EIBSM genotype k-3358. Most likely, this response was associated with the high-EIBSM trait, and was featured with pronounced enhancement in expression of the proteins related to redox metabolism. Thereby, *L*-ascorbate oxidase appeared to be among the proteins, which responded to NB+AMF inoculation with a pronounced increase in relative abundance. Most likely, this protein is involved in stress tolerance mechanisms [59,60]. Earlier, suppression of this gene was shown in senescent nodules of *Lotus japonicas* [61]. Probably, abundance of this protein indicates the active status of the root nodules of k-8274 upon combined inoculation with NB+AMF. On the other hand, the low-EIBSM genotype k-3358 responded to NB+AMF inoculation with suppression of root protein metabolism and enhancement of stress response, manifested as increased levels of phenolic/lignin biosynthesis. These findings should be verified in further experiments targeting other post-genomic aspects, i.e., at the transcriptome, metabolome and enzyme activity levels. The data on the EIBSM-responsivity at the level of the root proteome were in a good agreement with the plant response to mineral nitrogen salts supplemented in the soil. This response was much more strongly manifested in the roots of the k-3358 plants, which showed an increase in abundance of a ferredoxin–nitrite reductase, probably reflecting the activation of available nitrate assimilation. In general, our results will be helpful for further breeding work aimed at creation of new legume varieties with high responsivity to inoculation with beneficial soil microorganisms.

## Figures and Tables

**Figure 1 ijms-26-00463-f001:**
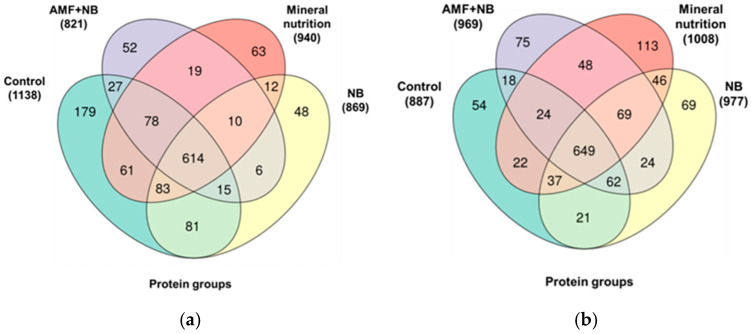
The numbers of non-redundant proteins (protein groups), identified in the *P. sativum* genotypes k-8274 (**a**) and k-3358 (**b**) plants grown in the absence of soil supplementations (controls), plants grown with supplementation of mineral nitrogen salts in the absence of microorganism complementation (mineral nutrition, MN), plants inoculated with rhizobia (nodule bacteria, NB), and plants inoculated with rhizobia and arbuscular mycorrhizal fungi (NB+AMF).

**Figure 2 ijms-26-00463-f002:**
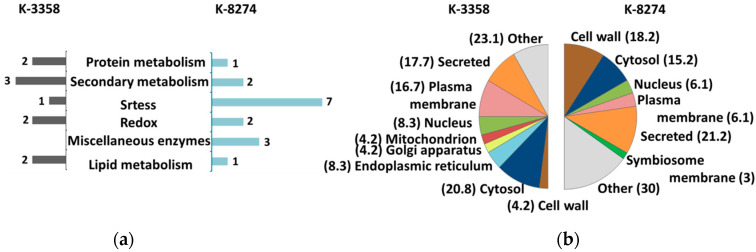
The most represented functional groups (**a**) and prediction of sub-cellular localization (**b**) for the proteins identified as differentially abundant in the roots of the k-3358 and k-8274 *P. sativum* plants. The individual proteins comprising each functional group (as well as all related information) are listed in Appendix A.

**Figure 3 ijms-26-00463-f003:**
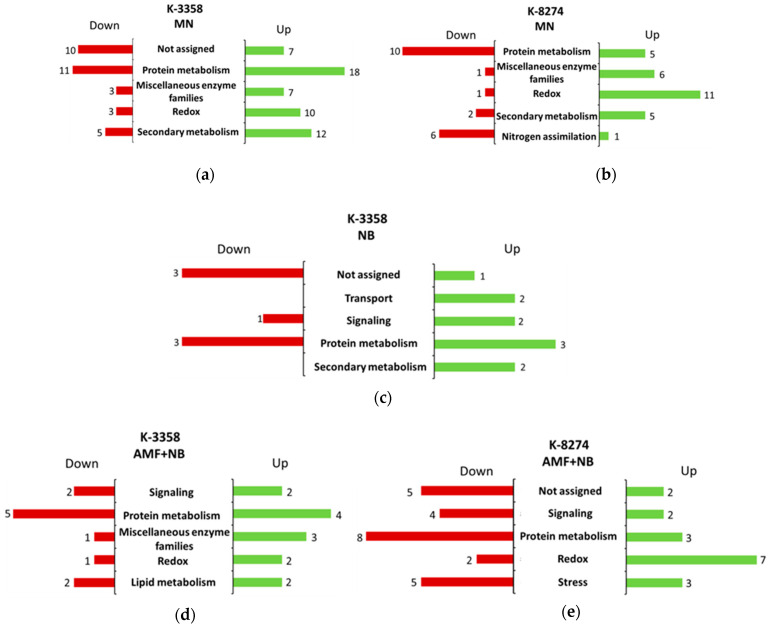
Top five functional groups (bins) representing the proteins identified as differentially abundant in *P.sativum* genotypes k-3358 (**a**,**c**,**d**) and k-8274 (**b**,**e**) under different soil supplementation conditions—external mineral nitrogen nutrition (MN, (**a**,**b**)), symbiotic rhizobia (nodule bacteria—NB, (**c**), combined complementation of arbuscular mycorrhiza and nodule bacteria (AMF+NB, (**d**,**e**)). Numerical values indicate the numbers of proteins constituting individual up- (green) or down-regulated (red) functional classes. The individual proteins comprising each functional group (in addition to all related information) are listed in Appendix A.

**Figure 4 ijms-26-00463-f004:**
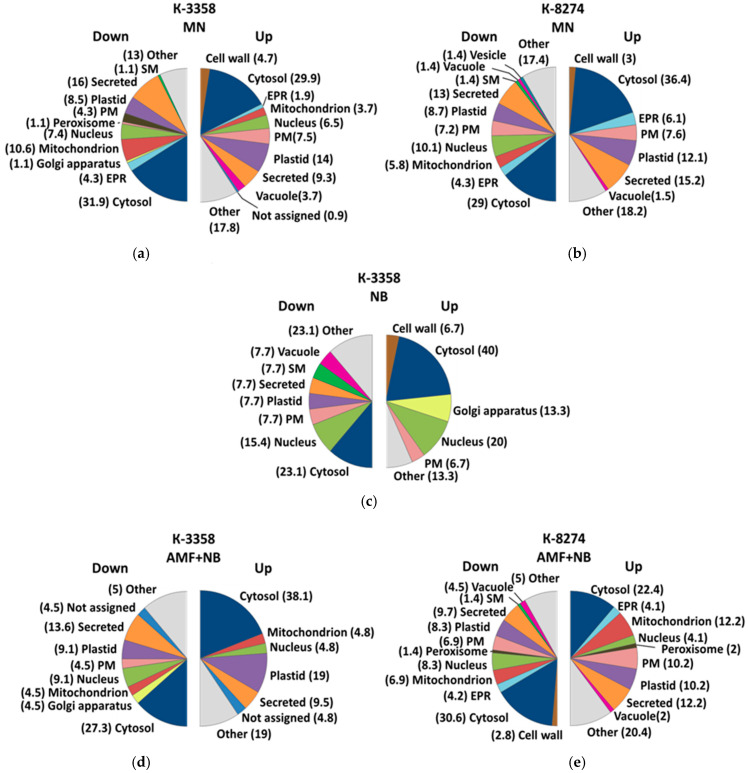
Prediction of the sub-cellular localization of proteins identified as up- and down-regulated in *P. sativum* genotypes k-3358 (panels (**a**,**c**,**d**)) and K 8274 (**b**,**e**) under external mineral nitrogen nutrition conditions (MN, panels (**a**,**b**)), upon inoculation with nodule bacteria (NB, panel (**c**)), upon the combined inoculation with arbuscular mycorrhiza and nodule bacteria (AMF+NB, panels (**d**,**e**)). The individual proteins annotated to specific predicted compartments are listed in Appendix A. SM—symbiosome membrane, EPR—endoplasmic reticulum, PM—plasma membrane.

**Table 1 ijms-26-00463-t001:** The most strongly regulated differentially expressed proteins with the highest impact on the inter-group differences observed in the paired inter-group comparisons in terms of the label-free quantification (LFQ) approach.

№	Description	Accession ^a^	Direction of Alterations ^b^	FC ^c^	*p_adj_* ^d^	Prediction of Localization	Functional Annotation
	Control of k-8274 in comparison with control of k-3358						
1	Dirigent protein	Psat7g248720.1	up	5.3	1.1 × 10^−2^	Sc	Secondary metabolism
2	Data Histone H2B	Psat0s3083g0040.1	up	3.7	3.8 × 10^−3^	Nc	DNA metabolism
3	Membrane steroid-binding protein 2-like	Psat6g199920.1	up	3.6	5.7 × 10^−3^	PM	Redox
4	HTH myb-type domain-containing protein	Psat5g257320.1	down	16	2.8 × 10^−2^	Nc	RNA metabolism
5	Phospholipase D	Psat5g302040.1	down	6.4	2.5 × 10^−3^	Ct, PM, Pl, EPR, Mt, Nc, Pd, Vc	Lipid metabolism
6	Carboxypeptidase	Psat1g016320.1	down	4.1	3.9 × 10^−3^	Sc	Protein metabolism
	k-3358: Control in comparison with mineral nutrition						
7	Histone H2B	Psat0s3083g0040.1	up	11.4	2.5 × 10^−5^	Nc	DNA metabolism
8	Non-symbiotic hemoglobin	Psat7g205800.1	up	4.3	1.8 × 10^−2^	Ct, CW, Pd, PM	Redox
9	Ferredoxin-nitrite reductase	Psat7g123960.1	up	4.2	2.0 × 10^−3^	Pl	Nitrogen assimilation
10	Heat shock protein DnaJ	Psat5g156720.1	down	1390.8	2.5 × 10^−5^	Pl	Stress
11	Leghemoglobin K	Psat7g013000.1	down	150.9	5.4 × 10^−5^	Ct	Nitrogen assimilation
12	Carbonic anhydrase	Psat0s2720g0080.1	down	129.2	7.8 × 10^−4^	Mt	Minor carbohydrate metabolism
	k-8274: Control in comparison with mineral nutrition						
13	Putative L-ascorbate oxidase	Psat4g070480.1	up	6.4	3.7 × 10^−2^	Sc, CW, Vc, PM, Pd	Development
14	Lipoxygenase	Psat0s1212g0080.1	up	4.1	2.5 × 10^−2^	Ct	Hormones
15	Ferredoxinnitrite reductase	Psat7g123960.1	up	3.3	1.6 × 10^−2^	Pl	N-metabolism
16	Beta-fructofuranosidase cell wall isozyme 2-like protein	Psat4g188080.1	down	246.3	2.1 × 10^−3^	Sc	Major carbohydrate metabolism, Minor carbohydrate metabolism
17	Putative chromatin regulator PHD family	Psat5g278840.1	down	121.1	9.6 × 10^−5^	Nc	Protein metabolism
18	Early noduline 7	Psat0s240g0040.1	down	85.5	6.3 × 10^−4^	SM	Nitrogen assimalation
	k-3358: Control in comparison with inoculation by nodule bacteria						
19	60S ribosomal protein L5-2	Psat0s1164g0040.1	up	3.9	2.5 × 10^−2^	Ct	Protein metabolism
20	Glutelin type-B-like protein	Psat7g061840.2	up	2.5	2.1 × 10^−2^	Ct	Protein metabolism
21	Phenylalanine ammonia-lyase	Psat6g072360.1	up	2.5	1.7 × 10^−2^	Ct	Secondary metabolism
22	CBS/octicosapeptide/phox/Bemp1 (PB1) domain protein	Psat5g058880.1	down	16.4	2.1 × 10^−2^	Pl, Mt	Amino acid metabolism, Protein metabolism
23	Putative serine/threonine-protein kinase	Psat1g036280.1	down	2.4	2.5 × 10^−2^	PM	Protein metabolism
24	Leghemoglobin	Psat0s241g0080.1	down	2.3	2.0 × 10^−2^	Ct	Nitrogen assimalation
	k-3358: Control in comparison with inoculation by arbuscular micorrhiza and nodule bacteria						
25	Caffeic acid O-methyltransferase	Psat3g198600.1	up	2.8	1.6 × 10^−2^	Ct, Nc, Pd, Pl	Secondary metabolism
26	Histone H2B	Psat0s3083g0040.1	up	2.5	8.2 × 10^−3^	Nc	DNA metabolism
27	Leucine-tRNA ligase, cytoplasmic isoform X1	Psat5g170080.1	up	2.4	4.0 × 10^−2^	Ct	Protein metabolism
28	Putative serine/threonine-protein kinase	Psat1g036280.1	down	3.6	8.2 × 10^−3^	PM	Protein metabolism
29	Carboxypeptidase	Psat1g016320.1	down	3.3	8.2 × 10^−3^	Sc	Protein metabolism
30	Fatty acyl-CoA synthetase family protein	Psat7g253640.1	down	2.9	8.2 × 10^−3^	Ct, Nc	Lipid metabolism
	k-8274: Control in comparison with mineral nutrition						
31	Monocopper oxidase SKS1-like protein	Psat4g070480.1	up	5.5	1.0 × 10^−2^	Sc, CW	Development
32	HTH myb-type Domain-containing protein	Psat5g257320.1	up	4.2	1.4 × 10^−2^	Nc	RNA metabolism
33	Arginase 1	Psat0s13993g0040.1	up	3.4	2.7 × 10^−2^	Mt	Amino acid metabolism, Stress
34	Heat shock protein DnaJ	Psat5g156720.1	down	67.8	4.4 × 10^−3^	Mt	Stress
35	Early nodulin ENOD18	Psat3g134480.1	down	44.9	9.5 × 10^−3^	Ct	Nitrogen assimalation
36	Putative serine/threonine-protein kinase	Psat1g036280.1	down	26.3	2.9 × 10^−3^	PM	Protein metabolism

^a^ Protein annotation relied on a combined sequence database, designed FASTA files of *Lotus japonicus*, *Medicago truncatula* and *Pisum sativum* from UniProtKB, combined into a single file (*Pisum sativum* V1r port) as described by Mamontova et al [19]; ^b^ “up” and “down” stands for up- and down-regulated proteins, respectively (i.e., those which increased and decreased their abunfdances upon soil supplementation with mineral salts of microorganisms; ^c^ FC, relative abundance fold change; ^d^
*p*_adj_, FDR (false discovery rate)-corrected probability values at the confidence level *p* ≤ 0.05 (correction was done in terms of the Progenesis QI LFQ protocol).

## Data Availability

The raw data are available in the PRIDE repository under the project accession number PXD058701 and project DOI 10.6019/PXD058701. Other information (i.e., any processed data/software outputs) are available upon request from the authors.

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
