# Peer review of "Responsivity of Two Pea Genotypes to the Symbiosis with Rhizobia and Arbuscular Mycorrhiza Fungi—A Proteomics Aspect of the “Efficiency of Interactions with Beneficial Soil Microorganisms” Trait"

_ijms, 2025, doi:10.3390/ijms26020463_

Round 1

Reviewer 1 Report

Comments and Suggestions for Authors

This study focuses on characterizing the alterations in the root proteome of highly responsive pea genotype K-8274 and low-responsive genotype K-3358 grown in non-sterile soil, which were associated with root colonization by rhizobia bacteria and arbuscular mycorrhizal fungi compared to proteome shifts caused by soil supplementation with mineral nitrogen salts. The results show that supplementation of the soil with mineral nitrogen-containing salts switched the root proteome of both genotypes to assimilate the available nitrogen, while processes associated with nitrogen fixation were suppressed. The most pronounced response was observed in the highly responsive K-8274 genotype inoculated simultaneously with rhizobial bacteria and arbuscular mycorrhizal fungi. This response involved activation of proteins related to redox metabolism and suppression of excessive nodule formation. In contrast, the low-responsive genotype K-3358 demonstrated a pronounced inoculation-induced suppression of protein metabolism and enhanced diverse defense reactions in pea roots under the same soil conditions. This is a well-designed research study. However, many details should be revised.

Line 144 should include a sentence indicating that the result was based on proteins selected for PCA.

Line 155 should show the 0-coordinate point for the x and y axes.

Line 222 should use genetic names such as K-8274 or K-3358 to clearly present these results.

Line 162 states that PCA accomplished for all samples obtained from the K-3358 plants revealed the separation of the four treatment groups with 53.4% and 13.4% of the total variance for PC1 and PC2, respectively. Where is the 2.1%?

Author Response

We thank the Reviewer for thoughtful review and highly appreciate the valuable comments and suggestions to improve the manuscript. Following these advices we performed all required changes in corresponding sections, as indicated in the following rebuttal addressing each aspect. We uploaded also all raw data in the PRIDE repository and provided appropriate data availability statement. The quality of language is checked with a native speaker.

All changes are marked in the text with blue and the corresponding remarks are specified with comments to the remarks of Reviewer 1.

Remarks

Remark 1

Line 144 should include a sentence indicating that the result was based on proteins selected for PCA

Answer: No, there was no selections in the experiments meant in the line 144. We are sorry for being not enough precise. We correct it here: “…which were applied to the whole annotated proteome”.

 Remark 2

Line 155 should show the 0-coordinate point for the x and y axes.

Answer: Unfortunately, we could not find the quantitative table to reproduce the figure 2. We don’t have time for new processing of the raw data (which are available now also in repository) we decided to remove the PCA part from the manuscript. All other quantitative data is represented in supplementary information and remains available. We are convinced, that this did not affect the quality of the manuscript and the conclusions done.

Remark 3

Line 222 should use genetic names such as K-8274 or K-3358 to clearly present these results.

Answer: We have added genetic names to make the text clearer

Remark 4

Line 162 states that PCA accomplished for all samples obtained from the K-3358 plants revealed the separation of the four treatment groups with 53.4% and 13.4% of the total variance for PC1 and PC2, respectively. Where is the 2.1%?

Answer: These 2.1% could be attributed tot he differences in other proncipal components, starting from PC3. We complemented the text to make it more clear: “…while the rest 2.1% could be attributed to further principal components”.

Reviewer 2 Report

Comments and Suggestions for Authors

please see paper for sticky notes

the paper could be edited for better reading

but the work is in an area that has wanted sound molecular analyses for a long time

can you add data on the biology of the response- you know the outcomes the reader does not

a summary diagram would be useful

why do you think the two cultivars  have these differences

suggestions are made to shorten the extensive lists of enzymes  /proteins that change    and putting them into functional groups  ie  it would help to do more discussion earlier

the supplemental data were not available  

for me the discussion v  results ratio were not optimal  

any inferences of what might be happening with P?

discussion of phenolics as signalling molecules  gating the processes of recognition and then symbiosis      again  summary diagrams useful 

Comments on the Quality of English Language

the paper requires an edit    and i think simplification of data sets  otherwise there is a huge overload   of facts   

Author Response

We thank the Reviewer for thoughtful review and highly appreciate the valuable comments and suggestions to improve the manuscript. Following these advices we performed all required changes in corresponding sections, as indicated in the following rebuttal addressing each aspect. We uploaded also all raw data in the PRIDE repository and provided appropriate data availability statement. The quality of language is checked with a native speaker.

All changes are marked in the text with green and the corresponding remarks are specified with comments to the remarks of Reviewer 2.

Remarks

General

General remark A

can you add data on the biology of the response- you know the outcomes the reader does not

Answer: We are absolutely agree with the reviewer - we extended the beginning of the Discussion part and added some relevant information- In the referred publication (Afonin et al) our earlier biological data is provided

General remark B

a summary diagram would be useful

Answer: We absolutely agree with the reviewer! Moreover, we are convinced that it must be a graphical abstract. So, now such a summary figure - a graphical abstract, it is provided

General remark C

why do you think the two cultivars  have these differences

Answer: we added the explanation to the Conclusion part. We believe that these differences reflect different ecological strategies of plants that can be more or less beneficial depending on the environmental conditions and/or biotic/abiotic stress factors.

General remark D

suggestions are made to shorten the extensive lists of enzymes  /proteins that change    and putting them into functional groups  ie  it would help to do more discussion earlier

Answer: We agree with the reviewer - it would be much more convenient. Please, see Supplementary information 4 for all proteins functional groups summarized

General remark E

the supplemental data were not available  

Answer: We checked it - now supplementary information provided

General remark F

for me the discussion v  results ratio were not optimal 

Answer: The Discussion part is extended as suggested by the Reviewer

General remark G 

any inferences of what might be happening with P?

Answer: We did not detect signs of the phosphorous starvation in any of the studied proteomes, so probably the amount of the phosphorus in the soil was sufficient for pea plants (note that this non-sterile soil contained indigenous microorganisms, including many rhizobial strains and many AM fungal isolates). Therefore we think that the effect of inoculation, at least at the sample collection stage (4 weeks after planting) may be connected with induction of plant defense rather that with nitrogen and/or phosphorus supply of plants with microsymbionts.

General remark H

discussion of phenolics as signalling molecules  gating the processes of recognition and then symbiosis      again  summary diagrams useful

Answer: We added some  information to the Discussion section, particularly about high- and low-EIBSM pea genotypes that differ in expression of genes related to flavonoid biosynthesis.

Other remarks

Remark 1

Replace “we decided for” with “we used”

Answer

Changed accordingly

Remark 2

Remove lines 87 - 89

Answer

Changed accordingly

Remark 3

Don’t understand explain what you mean by group

Answer

Here we mean the dispersion within one treatment group – i.e. dispersion within one group of plants grown under the same soil conditions. The appropriate text is provided: “i.e. dispersion within one group of plants grown under the same soil conditions”

Remark 4

Replace “could be” with were

Answer

Changed accordingly

Remark 5

Provide as Venn diagrams with all groups

Answer

We agree with the reviewer, such presentation is quite helpful - the corresponding Venn diagrams are presented now as Figure 1, S1-3 and S1-4

Remark 6

Change to italic

Answer

Changed accordingly

Remark 7

“Proteins” instead “protein”

Answer

Changed accordingly

Remark 8

You mean noduline 7 or what?

Answer

We mean here annotation as it given by the database search –“early noduline 7”

Remark 9

Replace “thereby” with “the”

Answer

Changed accordingly

Remark 10

Check throughout for unneeded  - b etween words   several of them

Answer

Changed accordingly

Remark 11

Replace with “used”

Answer

Changed accordingly- the text is changed: “For the functional annotation of the differentially expressed P. sativum proteome Mercator 4 v.2 software was used.”

Remark 12

Bins of??

Answer

Here we mean the word “bins”

Remark 13

Could these long lists be put into a table-to compare contrast between bacterial types-then  the text could be used to express highlights of the findings   

Answer

We are absolutely agree with the Reviewer. The requested tables are provided as Supplementary  materials 3 and 4- Due to their comprehensive character we decided for supplementary, but not the main text

Remark 14

ie its the whole cell that is responding

not unexpected though since many  metabolites nip in and out of organelles etc

Answer

We thank the Reviewer - this consideration is incorporated- The text is provided: “The results demonstrated that the whole cell that responded due to strong metabolic interconnection of the organelles.”

Remark 15

What is missing in text for me is a comparison table of the extent of nodulations and the amount of AMF colonization. There should be some discussion of potential  suppression or activation of host defense responses. Rhizobia and AMF  both   will enhance P  in the plant too. See Jeff Dangls review of how P levels affect defence responses

Answer

Unfortunately, no data on mycorrhization intensity were obtained in the experiment, therefore we can only hypothesize about the suppression or activation of the host defense responses due to inoculation. We added the discussion of possible activation of plant defense priming in k-8274 plants due to inoculation with AM, and cited and discussed several additional papers, including the review of Jeffery Dangl.

Remark 16

Specify the source of mineral nitrogen

Answer

Mineral nitrogen was used as ammonia nitrate - see now specified in the text

Remark 17

Please add data of nodulation  and colonization

Answer

The data on nodulation has already been published (Zhukov et al., 2017; we cite this paper several times). The data on mycorrhization were not obtained in that experiment due to technical reasons.

Remark 18

Check spelling throughout

Answer

Spelling is checked
